# Pilot plant study on nitrogen and phosphorus removal in marine wastewater by marine sediment with sequencing batch reactor

**Jinsoo Kim**[1], **Sangrim Kang**[2], **Hyun-Sook Kim**[2], **Sungchul Kim**[3]*, **Sang-Seob Lee**[1,2]*

**1** Department of Life Science, Graduate School, Kyonggi University, Suwon-si, Gyeonggi-do, Republic of Korea, **2** Department of Biological Engineering, Graduate School, Kyonggi University, Suwon-si, Gyeonggi-do, Republic of Korea, **3** Department of Environmental Energy Engineering, Graduate School, Kyonggi University, Suwon-si, Gyeonggi-do, Republic of Korea

* mac40@hanmail.net (SK); sslee@kyonggi.ac.kr (SL)

**Data Availability Statement:** All relevant data are within the manuscript and its Supporting Information files.

## Abstract

Effective biological treatment of marine wastewater is not well-known. Accumulation of nitrogen and phosphorus from land-based effluent is a crucial cause of red-tide in marine systems. The purpose of the study is to reduce nitrogen and phosphorus in marine wastewater with a pilot plant-scale sequencing batch reactor (SBR) system by using marine sediment as eco-friendly and effective biological materials, and elucidate which bacterial strains in sludge from marine sediment influence the performance of SBR. By applying eco-friendly high efficiency marine sludge (eco-HEMS), the treatment performance was 15 $m^3$ $d^{-1}$ of treatment amount in 4.5 $m^3$ of the reactor with the average removal efficiency of 89.3% for total nitrogen and 94.9% for total phosphorus at the optimal operation condition in summer. Moreover, the average removal efficiency was 84.0% for total nitrogen and 88.3% for total phosphorus in winter although biological treatment efficiency in winter is generally lower due to bacterial lower activity. These results were revealed by the DNA barcoding analysis of 16s rRNA amplicon sequencing of samples from the sludge in winter. The comparative analysis of the bacterial community composition in sludge at the high efficiency of the system showed the predominant genera *Psychromonas* (significantly increased to 45.6% relative abundance), *Vibrio* (13.3%), *Gaetbulibacter* (5.7%), and *Psychroserpens* (4.3%) in the 4 week adaptation after adding marine sediment, suggesting that those predominant bacteria influenced the treatment performance in winter.

## Introduction

Nitrogen (N) and phosphorus (P) released from land-based effluents is believed to be one reason which have increased N and P in the ocean [1]. Sequentially, accumulation of these nutrients could cause the harmful algal blooms (HABs) in marine coastal regions [2]. HABs as a pollution source bring out imbalanced marine ecosystems despite marine environmental precautions and protection efforts [2–5]. When HABs occur regularly in fishery areas, fishery production, marine ranching, and aquaculture in coastal or inland areas must be completely

**Funding:** This work was supported by the National Research Foundation of Korea under Grant NRF-2015M3A9B8029697 and NRF-2017M3A9B8065734. Marine bacteria KGN1 (KEMB 3401-006) and KGP1 (KEMB 3001-129) were obtained from the Korea Environmental Microorganisms Bank. Jinsoo Kim and Sungchul Kim are employed by and receive salary from Research & Development Institute of Inventory Co. Ltd.

**Competing interests:** Jinsoo Kim and Sungchul Kim are employed by and receive salary from Research & Development Institute of Inventory Co. Ltd. The specific roles of these authors are articulated in the 'author contributions' section. This does not alter our adherence to PLOS ONE policies on sharing data and materials.

protected to prevent economic loss in the fishery and marine industries [6, 7]. In recent decades, scientific activities and government monitoring in South Korea have found that increasing nitrogen and phosphorus contents in seawater quality of coastal areas from the impact of inland pollution sources [8, 9]. Therefore, treatment of effluent from fish farms and coastal area is important in a long-term the prevention strategy.

However, treatment of marine wastewater needs better understood because the high salinity of wastewater from land-based fish farms and fishery production facilities in the coastal area has hampered the formulation of an effective solution for biological treatment of wastewater [2, 10, 11]. For the reasons, physio-chemical methodologies such as ultraviolet irradiation/ozone, reverse osmosis, ion exchange, electro-dialysis, and photo catalysis have been applied to the marine wastewater, but they are very expensive and have caused secondary problems such occurrence of biofilm and derived chemicals [12, 13]. Thus, biological treatments have been proposed for more effective and safety methodology to remove nutrient salts in the high salinity of wastewater. The development of a sludge should have been preferred for effective application of biological method, because sludge should settle in highly saline water since marine wastewater can decrease settling of sludge that consists of various microorganisms by changing the bacterial community and lowering bacterial metabolism. Bacterial cell lysis occurs in unaccustomed bacteria in sludge to marine wastewater because of the osmotic difference in the treatment of marine wastewater [14–16]. Recent studies of the effective treatment of saline wastewater have utilized halo-tolerant marine bacteria, such as *Halomonas* sp., *Aeromonas* sp., *Bacillus* sp., *Staphylococcus xylosus*, and *Vibrio diabolicus* as the predominant microorganism. Our previous study found that *Bacillus* sp. KGN1 (KEMB 3401–006) efficiently removed nitrogen (N) with 86.0% removal efficiency (RE) for 10 h at initial 10 mg $L^{-1}$ $NH_3$-N and *Vibrio* sp. KGP1 (KEMB 3001–129) efficiently removed phosphorus (P) with 99.9% RE for 10 h at initial 10 mg/L $PO_4^{3-}$P [17]. Moreover, aerobic granular sludge (AGS) instead of common activated sludge bacteria was used for improving settlement in the biological treatment of marine wastewater [10, 11, 17–19]. Thus, we also previously applied AGS to lab-scale SBR reactor, and found that AGS increased settlement of activated sludge for better efficiency in the biological treatment [18].

However, those studies showed the highly efficient performance in only the laboratory scale. The practitioners, managers and governors in marine industry always are longing to apply the economic solution to the field sites with more treatment amount. Meanwhile, biological materials also need more verification for well-adapted and economic value for the application to the field site. Thus, it is also required to study possible application of eco-friendly materials such marine sediment to the biological treatment. Moreover, understanding the predominant bacteria in the sludge is important to operate and manage for improving the performance of biological treatment of marine wastewater treatment. The nitrogen removal in wastewater treatment has well studied with nitrification bacteria such as ammonia-oxidizing bacteria (AOB), anaerobic ammonium oxidation (anammox) bacteria and denitrification bacteria [20–22]. Moreover, recent studies reported that phosphorus in wastewater treatment with high salinity was removed by denitrifying phosphorus removal (DPR) bacteria and phosphorus uptake metabolism [23, 24]. Thus, DNA barcoding has known as an useful metagenomics tool for how those effective bacteria in the bacterial community in sludge influenced and improved the treatment of marine wastewater [25].

In this study, we aim 1) to remove nitrogen and phosphorus in marine wastewater from a land based fish farm by marine sediment with a pilot plant-scale SBR; 2) to analyze the bacterial communities during adaptation of marine sediment to marine sludge (eco-HEMS). We hypothesize that the effective bacteria of the eco-HEMS improves the nitrogen and phosphorus

removal and treatment performance of marine wastewater as bacteria adapted in the saline wastewater.

## Materials and methods

### Bacteria and eco-friendly high efficiency marine sludge (eco-HEMS)

Marine sediment in a volume of 0.5 m$^3$ was collected from Jebudo Island, Hwaseong-si, South Korea (37˚09'45.3" N, 126˚37'19.5" E, S1(a) and S1(c) Fig) and sieved with 10 mesh (0.55 mm pore size) after removing tiny stones and debris. The sieved marine sediment was applied to a sequencing batch reactor (SBR) system in pilot plant-scale with SBR cycles (more description in the next section). Bacterial strains used in this study were described in our previous study [17], whereas *Bacillus* sp. KGN1 (KEMB 3401–006) and *Vibrio* sp. KGP1 (KEMB 3001–129) efficiently removes nitrogen and phosphorus. Those bacterial strains were routinely maintained and cultured in Difco 2216 marine broth (BD, Franklin Lakes, NJ, USA). Two bacterial strains were bulk-cultivated for 5 days in a 1.5 m$^3$ bio-reactor (Nexus Co. Ltd., Seoul, South Korea) at a maintained temperature of 28˚C and 1.5–2.0 mg L$^{-1}$ of dissolved oxygen (DO), and then bacterial cells were collected with the high-speed tubular separator (Hanil SME Co. Ltd,, Seoul, South Korea) at 20,000 × g and a flow rate of 600 L h$^{-1}$. In 2 week after adding marine sediment, the bacterial two strains pellets in 1 kg (wet weight) were added to the adapted marine sediment. During adaptation period for 2 weeks, marine sediment was maintained in an SBR cycle with a supplement of D-glucose, NH$_4$Cl, and KH$_2$PO$_4$, and adjusted to a chemical oxygen demand of 100 mg L$^{-1}$ by the dichromate method (COD$_{Cr}$) in the presence of 5 mg L$^{-1}$ NH$_3$-N and 1 mg L$^{-1}$ PO$_4$$^{-3}$-P in the marine wastewater. The SBR cycling comprised four steps (influence/mix–aeration–settlement–idle/effluence). Meanwhile, aerobic granule sludge (AGS) in a volume of 1.0 m$^3$ was employed in another setting as the control because AGS has high settlement with high nitrogen removal efficiency. The mixed liquor suspended solids (MLSS) concentration was adjusted to 1,500 mg L$^{-1}$ for the setup since 1,500 mg L$^{-1}$ of MLSS showed the highest efficiencies in the previous lab-scale reactors study.

### Pilot plant-scale SBR system and operation conditions

The pilot plant-scale SBR biological system was designed as 10 m$^3$d$^{-1}$ of daily wastewater flow (Q) with 4.5 m$^3$ of SBR reactor in a total volume (V$_T$), 2.5 m$^3$ of filling volume (V$_F$), and 6 h cycle $^{-1}$ of time per cycle (T$_C$) from previously obtained parameters in lab-scale data [17, 18], and setup in Tongyeong-si, Gyeongsangnam-do, South Korea (34˚49'31.82"N 128˚20'10.17"E, S1(b) and S1(c) Fig). Tongyeong is located in the southern-coastal area of South Korea in which red tide frequently occurred according to the statistical data [26]. The optimal conditions obtained from the lab-scale reactor were applied to the pilot plant-scale SBR system at the setup.

The pilot-scale SBR biological system had been in operation for approximately 2 years (June 2015 to January 2017). Marine wastewater was used as the effluent from the land-based fish farms, Tongyeong. The system is depicted in Fig 1.

Marine wastewater in the wastewater storage tank flowed into the influence storage tank [1.2 m (W) × 1.7 m (L) × 1.8 m (H)] and then was pumped to the SBR reactor [2.0 m (W) × 1.7 m (L) × 1.8 m (H)], which was where the eco-HEMS sludge biologically reacted to remove nutrient salts in the marine wastewater. The treated water then was decanted to the effluent storage tank [1.2 m (W) × 1.7 m (L) × 1.8 m (H)] for external discharge. The pilot-scale SBR system was regulated with an auto-control system comprising an electronic touch panel placed in the container (3.0 m × 9.0 m). The SBR reactor is the single reactor with 4.5 m$^3$ of V$_T$ housing nine aerators and a decant system.

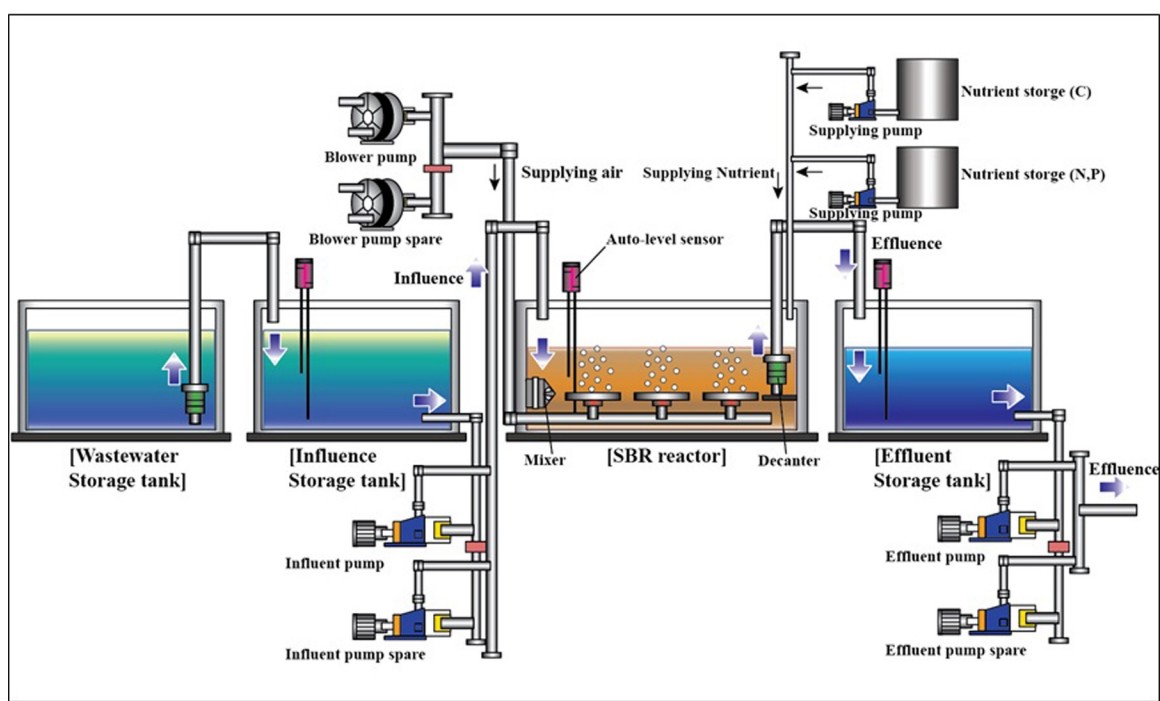

**Fig 1. Schematic diagram of the SBR biological treatment system for effluent from the land-based fish farm.**

Unlike expectation with 6 h cycle$^{-1}$ of $T_C$, eco-HEMS showed higher treatment performance, so the pilot plant-scale SBR operated 4 h cycle$^{-1}$ of $T_C$ with following four stages: the influence stage (0.5 h) within the aeration/ mixing stage (3.0 h), settlement stage (0.5 h), and idle/effluence stage (0.5 h). The system operated with 7.2 h hydraulic retention time (HRT), 24.5 d solids retention time (SRT) and exchange ratio ($V_F/V_T$) of 0.56, since the filling volume ($V_F$) was 2.5 m$^3$ cycle$^{-1}$. As reducing reaction time, Q was 15 m$^3$ d$^{-1}$. Based on previous studies, MLSS was adjusted and routinely maintained at 1,500 mg L$^{-1}$, and the mixed liquor volatile suspended solids (MLVSS) were adjusted and routinely maintained at approximately 1,200 mg L$^{-1}$ (Table 1).

**Table 1. Optimal operation conditions of pilot plant-scale SBR biological treatment for marine wastewater with eco-HEMS during operation period (2 months, n = 360 cycles).**

| Operation conditions | eco-HEMS | AGS |
|---|---|---|
| Q (m$^3$ d$^{-1}$) | 15.0 | 10.0 |
| $V_T$ (m$^3$) | 4.5 | 4.5 |
| $V_F$ (m$^3$) | 2.5 | 2.5 |
| $T_C$ (h • cycle$^{-1}$) | 4.0 | 6.0 |
| Cycles • d$^{-1}$ | 6.0 | 4.0 |
| HRT(h) | 7.2 | 10.8 |
| SRT (d)[a] | 24.5 | 25.0 |
| F/M (g COD • g MLSS$^{-1}$ d$^{-1}$) | 0.278 | 0.185 |
| MLSS (mg L$^{-1}$) | 1,500 | 1,500 |
| MLVSS (mg L$^{-1}$) | 1,200 | 1,200 |

* a, SRT maintained 24.5 d in the winter, but 20.0 d in the summer (AGS also maintained 20.0 d in the summer).

Although the nutrient quality in the marine wastewater varied, the influence was adjusted with the daily supplementation of D-glucose (acetate for AGS), $NH_4Cl$, and $KH_2PO_4$ depending on the purpose of the study. During the operating period, profiling of eco-HEMS and AGS was done in the pilot-scale SBR system at various $COD_{Cr}$: N: P (below C: N: P) ratios. Data were collected in all seasons.

## Analysis of environmental factors, nutrients, and kinetic parameters

During the operation periods, analysis of chemicals including the MLSS was routinely performed. The environmental factors of marine wastewater such as temperature, salinity, DO, and pH were also routinely determined by each portable equipment. $COD_{Cr}$, MLSS, MLVSS, N, and P were analyzed as following the standard methods [27]. $COD_{Cr}$ analysis was carried out by soluble $COD_{Cr}$ after filtration. $COD_{Cr}$, ammonia N ($NH_3$-N), nitrate N ($NO_3$-N), total N (T-N), phosphate P ($PO_4^{3}$-P), and total P (T-P) were determined using a model DR4000 spectrophotometer (HACH Co., Frederick, MD, USA) followed as the HACH manual. The REs of $COD_{Cr}$, $NH_3$-N, and $PO_4^{3}$-P were quantified using the initial and final concentrations. For evaluation of sludge settlement, sludge volume in settlement for 30 min ($SV_{30}$) was determined as following the standard methods [27]. For the operational evaluation of the SBR system in the pilot plant-scale, analytical data were used to calculate the kinetic parameters of HRT, SRT, bacterial growth yield (Y), specific growth rate (μ), and solid volume index (SVI) using the appropriate formula. For comparative analysis of the efficiency between eco-HEMS and AGS, specific efficiency component per unit sludge (MLSS) was determined for $NH_3$-N, $NO_3$-N, and P removal according to reaction time in response to bacterial growth. Specific substrate utilization rate (SSR, Eq 1), specific nitrification rate (SNR, Eq 2), specific denitrification rate (SdNR, Eq 3), and specific phosphate uptake rate (SPR, Eq 4) were calculated using the measured concentrations as follows:

$$SSR(g\ COD_{Cr}\ Removal \cdot g\ cell^{-1} \cdot d^{-1}) = (S_0 - S)/(HRT \times (X - X_0)) \qquad (1)$$

$$SNR(g\ NH_3 - N\ Removal \cdot g\ cell^{-1} \cdot d^{-1}) = (N_0 - N)/(HRT \times (X - X_0)) \qquad (2)$$

$$SdNR(g\ NO_3^- - N\ Removal \cdot g\ cell^{-1} \cdot d^{-1}) = (dN_0 - dN)/(HRT \times (X - X_0)) \qquad (3)$$

$$SPR(g\ PO_4^{3-} - P\ Removal \cdot g\ cell^{-1} \cdot d^{-1}) = (P_0 - P)/(HRT \times (X - X_0)) \qquad (4)$$

Where $S_0$, is the initial $COD_{Cr}$ concentration; S is the final $COD_{Cr}$ concentration; $N_0$ is the initial $NH_3$-N concentration; N is the final $NH_3$-N concentration; $dN_0$ is the initial $NO_3^-$-N concentration; dN is the final $NO_3^-$-N concentration; $P_0$ is the initial $PO_4^{3-}$-P concentration; P is the final $PO_4^{3-}$-P concentration; $X_0$ is the initial MLSS; and X is the final MLSS.

## Analysis of bacterial community

Each 100 mL of sludge sampled from the 10 sites of the SBR reactor was collected and mixed, and then 50 mL of them was used for every bacterial community analysis. Sampling was performed every week after the marine sediment had adapted to the eco-HEMS. The bacterial community obtained in the summer and winter using the optimal operating conditions were compared. Comparative analysis of the bacterial community of AGS in the summer was performed. All bacterial communities were analyzed by DNA barcoding sequencing outsourced to Macrogen Inc. (Seoul, South Korea). Sequencing was performed on the Miseq 15027617 system with V4 region in 16S ribosomal RNA (Illumina Inc., San Diego, CA, USA) and the

generated raw images were analyzed MiSeq Control Software v2.2 for system control and base calling through Real Time Analysis. v1.18 integrated primary analysis software. The base calls binary was converted into FASTQ utilizing the bcl2fastq (v1.8.4) package (Illumina). Adapters were trimmed away from the reads [28]. Taxonomic assignment of the sequenced reads was carried out using previously described methodology [18]. From the obtained data, a statistical analysis of the bacterial community composition was weekly performed during marine sludge adapted Eco-HEMS sludge from the beginning to fifth week. The relative difference in bacterial composition during the activation of the marine sediment was analyzed by principle component analysis (PCoA) to evaluate the bacterial community shift.

## Results and discussion

### 1. Performance of the SBR biological treatment system with eco-HEMS

The pilot plant-scale study was performed to scale up the treatment volume with high efficiency and to provide a system that would be useful for fish aquaculture managers and practitioners using effluent from the land-based fish farms. Daily wastewater treatment amount (Q) is important. The SBR biological treatment system was designed and setup with a Q value of 10.0 m$^3$ d$^{-1}$ based on the data of the lab-scale study. Unexpectedly, the Q of the pilot plant-scale SBR biological treatment with eco-HEMS increased from 10.0 m$^3$ d$^{-1}$ to 15.0 m$^3$ d$^{-1}$, since the reaction time in the SBR cycle shorten from 6 h cycle$^{-1}$ to 4 h cycle$^{-1}$ (Table 1). As a result, HRT could decrease to 7.2 h. SRT was maintained and operated for 20 d, except in winter when it increased to 24.5 d.

**Environmental factors and seasonal performance.** During the operation period, the environmental factors of marine wastewater from land-based fish farms were investigated. These included temperature, pH, and salinity, which were important parameters to be considered when operating and maintaining the activated sludge (both AGS and eco-HEMS) in the pilot plant-scale SBR system. Trends of temperature and salinity showed seasonal fluctuation, but there was no significant difference in seasonal pH trends on average (Table 2 and S2 Fig).

From monitoring results of environmental factors, the spring and autumn temperatures were similar. Interestingly, seasonal salinity decreased from spring to summer and autumn, then increased again from winter to spring. Salinity in summer was similar to that in autumn, and salinity in winter was similar to that in spring. The average salinity in spring and winter was lower (average 2.2–2.7 PSU) than that in winter and autumn. The pH was highest in autumn and the pH difference was greatest in summer. The pH and salinity values were due to the typhoon and rainy season that occurs in the summer and early autumn on the south coastal area of South Korea. The concentration of DO, which is another influential environmental factor, was not considered because the equalization tank of the SBR system takes role of storage and equalization of marine wastewater and air was supplied to the SBR tank of the system (Fig 1). From all data, the seasonal performance results of the pilot plant-scale SBR biological treatment indicated that eco-HEMS showed similar REs in both spring and autumn with on the average, 75.8% of COD$_{Cr}$ (initial: 149.0 mg L$^{-1}$), 54.9% of total nitrogen (T-N, initial: 9.3 mg L$^{-1}$), 65.5% of total phosphorus (T-P, initial: 1.8 mg L$^{-1}$). In summer, eco-HEMS showed, on the average REs, 51.2% of COD$_{Cr}$ (initial: 100.8 mg L$^{-1}$), 70.9% of T-N (initial: 5.9 mg L$^{-1}$), 47.9% of T-P (initial: 1.8 mg L$^{-1}$). Surprisingly, in winter, eco-HEMS showed higher average REs with 65.5% of COD$_{Cr}$ (initial: 158.0 mg L-1), 82.0% of T-N (initial: 9.7 mg L$^{-1}$), 79.7% of T-P (initial: 2.0 mg L$^{-1}$). Meanwhile, AGS showed lower average REs in winter with 67.4% of COD$_{Cr}$ (initial: 164.8 mg L$^{-1}$), 53.3% of T-N (initial: 6.7 mg L$^{-1}$), 52.5% of T-P (initial: 2.0 mg L$^{-1}$) (S1 Table and S3 Fig). COD$_{Cr}$ concentration in effluent was below 40.0 mg L$^{-1}$, T-N was below 5.0

**Table 2. Environmental factors of marine wastewater from land-based farming on each season during operation periods.** Spring, from March to May; Summer, June to 21 September; Autumn, from 22 September to November; Winter, from December to February.

| Environmental factor | Season | Spring | Summer | Autumn | Winter |
|---|---|---|---|---|---|
| pH | AVE | 7.8 | 7.7 | 8.0 | 7.7 |
| | min-MAX | 7.3–8.3 | 6.5–8.2 | 7.5–8.7 | 7.4–8.3 |
| Temperature (˚C) | AVE | 14.2 | 22.7 | 14.5 | 9.02 |
| | min-MAX | 8.0–21.0 | 19.7–27.0 | 10.0–22.3 | 2.0–13.0 |
| Salinity (PSU) | AVE | 33.6 | 31.4 | 31.2 | 33.9 |
| | min-MAX | 32.4–34.5 | 30.2–32.6 | 30.3–32.6 | 32.6–34.9 |

mg $L^{-1}$, and T-P was below 0.6 mg $L^{-1}$. Common sea water was analyzed as averagely 42.0–43.0 mg $L^{-1}$ of $COD_{Cr}$, 5.0 mg of T-N, and 0.5 mg $L^{-1}$ of T-P.

**1.2. Comparative analytical profile between eco-HEMS and AGS in the optimal conditions.** In winter, eco-HEMS showed higher performance in the pilot plant-scale SBR treatment syste. Thus, the profiling of eco-HEMS was performed in winter with 100: 5: 1, 200: 5: 1 and 300: 5: 1 of C: N: P ratios, respectively. The eco-HEMS profile in the SBR system displayed optimal treatment efficiency in the winter when the C: N: P was 100: 5: 1 (Fig 2).

During the profiling of eco-HEMS with nutrient removal, the temperature was not significantly changed, but DO and pH sharply decreased from 0 min to 75 min and increased again from 75 min to 180 min in SBR stage II (aeration and mixing). At the start of SBR stage III (settlement in anaerobic condition), pH was not change and DO spontaneously decreased. $COD_{Cr}$ (initial: 103.5 ± 2.0 mg $L^{-1}$) decreased in the first 60 min, with no change thereafter (final: 51.2 ± 5.9 mg $L^{-1}$). The concentration of MLSS increased, with 1,460 ± 40 mg $L^{-1}$ to 1,620 ± 20 mg $L^{-1}$ during the total profile time of 240 min. MLSS displayed a logarithmic increase from 0 min to 60 min, and slightly increased from 60 min to 240 min. T-N (initial: 7.1 ± 0.5 mg $L^{-1}$) constantly decreased from the initial time to 180 min in SBS stage II, and slightly decreased in the SBR stage III (final: 1.1 ± 0.5 mg $L^{-1}$). $NH_3$-N sharply decreased from the initial time to 180 min. $NH_3$-N decreased and $NO_3^-$-N simultaneously increased by nitrification from the initial time to 60 min, and $NO_3^-$-N decreased from 60 min to 240 min by denitrification (initial: 5.2 ± 0.3 mg $L^{-1}$, $NO_3^-$-N, 1.9 ± 0.4 mg $L^{-1}$; final: $NH_3$-N, 0.2 ± 0.3 mg $L^{-1}$; $NO_3^-$-N, 0.9 ± 0.6 mg $L^{-1}$). P readily decreased from the initial time to 30 min, steadily decreased at the end of SBR stage II, and slightly increased in SBR stage III (initial: T-P 1.1 ± 0.1 mg $L^{-1}$, $PO_4^{3-}$-P 0.8 ± 0.1 mg $L^{-1}$; final: T-P 0.1 ± 0.1 mg $L^{-1}$; $PO_4^{3-}$-P, 0.1 ± 0.1 mg $L^{-1}$). The profiles of eco-HEMS when the C: N: P ratio was 200: 5: 1 and 300: 5: 1 are shown in the S4 and S5 Figs. The initial $COD_{Cr}$ was higher and at the time when DO was zero became longer, and pH was also lower. The $COD_{Cr}$ was reduced from the initial time to the end of SBR stage II, and the MLSS increased from 0 min to 60 min, and slightly increased from 60 min to 240 min. As the initial $COD_{Cr}$ concentration was higher, the RE of T-N decreased and that of P slightly decreased. As the ratio of $COD_{Cr}$ in the C: N: P was higher, removal of $NH_3$-N was greater, but the removal of $NO_3^-$-N was increased more by the nitrification of ammonia. P steadily decreased from the initial time to the end. Overall, the condition when C: N: P was 100:5:1 was optimal because REs of $COD_{Cr}$, N, and P were higher with consumption of DO by microbial growth, metabolism, and activity. Moreover, the N treatment efficiency by eco-HEMS was highest because the efficiency of nitrification and denitrification was the highest with lower biomass production. The P treatment efficiency by eco-HEMS was also highest. Comparatively, at the same condition (C: N: P is 100: 5: 1) in winter, the profile of AGS

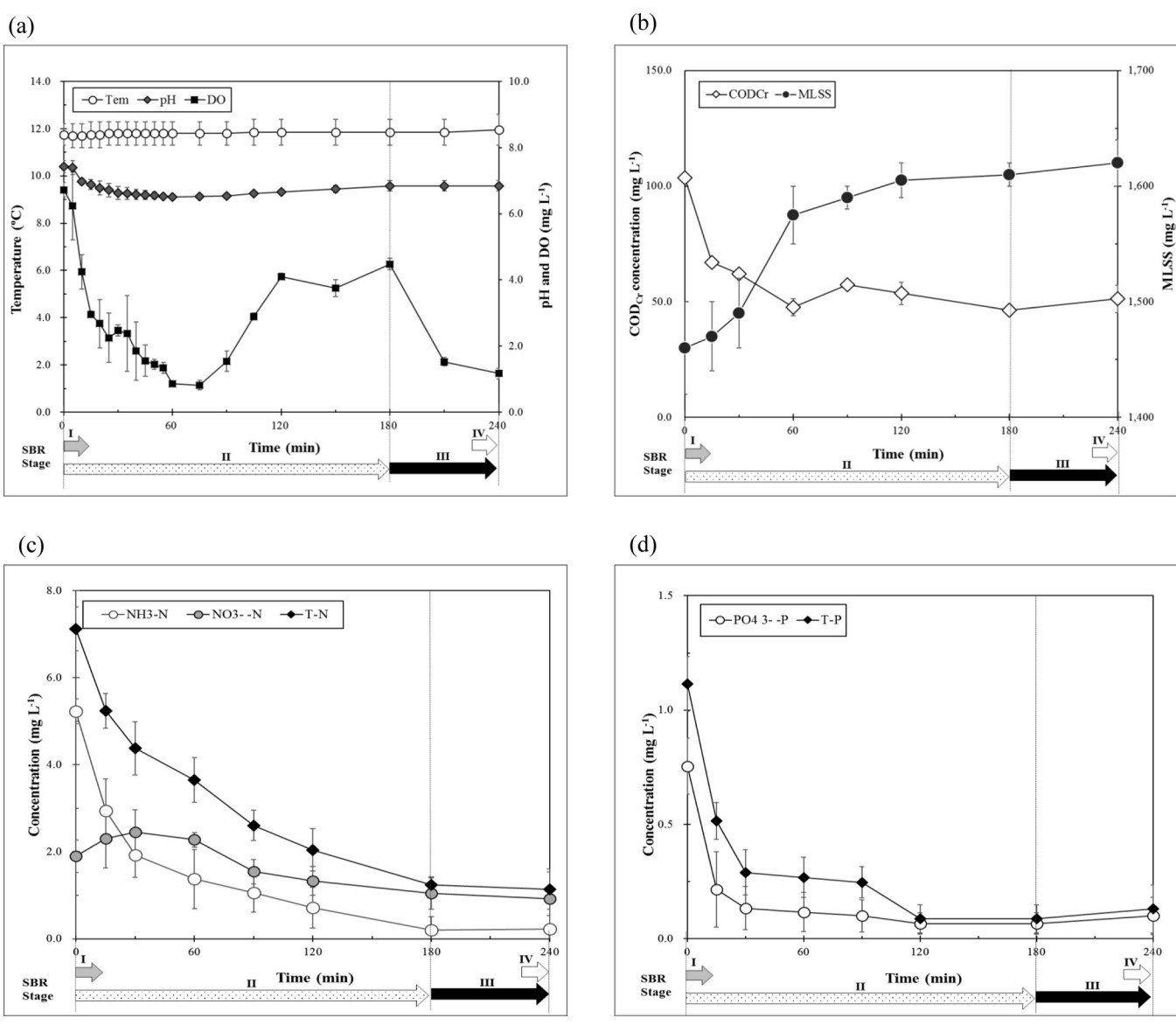

**Fig 2. Analytical profiles of eco-HEMS in the winter season, with COD:N:P of 100:5:1 as the optimal condition.** Environmental factors (a), $COD_{Cr}$ and MLSS (b), T-N, $NH_3$-N, and $NO_3^-$-N(c), and T-P and $PO_4^{3-}$-P(d).

revealed lower microbial activity and treatment efficiency of organic C, N, and P with lower DO consumption (Table 3).

AGS showed, on the average, REs of 31.8% for $COD_{Cr}$, 49.6% for T-N, 76.8% for $NH_3$-N, -46.8% for $NO_3^-$-N, 49.0% for T-P, and 45.2% for $PO_4^{3-}$-P. In the same condition, the profiles of eco-HEMS and AGS in the summer showed different trends in the S6 and S7 Figs and Table 3. The temperature was slightly changed (23 ± 2°C) in both conditions. DO steadily decreased from the initial time to the end of an SBR cycle (eco-HEMS: 4 h cycle$^{-1}$, AGS: 6 h cycle$^{-1}$), and pH trends were similar to that in winter, in which pH decreased from the initial time to 120 min in the profile of eco-HEMS (180 min in the profile of AGS during the time when $COD_{Cr}$ sharply decreased, pH increased again from 120 min to the end. The profile of eco-HEMS in the summer showed an averagely increase of 260 mg L$^{-1}$ for MLSS (increase of

**Table 3. Comparative results of $COD_{Cr}$, T-N, $NH_3$-N, $NO_3^-$ -N, T-P, $PO_4^{3-}$ -P Removal Efficiency (RE) between AGS and eco-HEMS in summer and winter season, when COD:N:P was 100:5:1 in the optimal condition.**

| Season | | Summer | | | | Winter | | | |
|---|---|---|---|---|---|---|---|---|---|
| Applying Sludge | | eco-HEMS | | AGS | | eco-HEMS | | AGS | |
| Reaction Time (h) in a cycle | | 0 | 4 | 0 | 6 | 0 | 4 | 0 | 6 |
| $COD_{Cr}$ (mg L$^{-1}$) | AVE | 110.9 | 40.1 | 122.5 | 46.0 | 103.5 | 51.2 | 111.8 | 76.3 |
| | STD | 10.4 | 1.7 | 9.1 | 9.6 | 2.0 | 5.9 | 9.5 | 10.2 |
| | RE (%) | | 63.9 | | 62.5 | | 50.6 | | 31.8 |
| T-N (mg L$^{-1}$) | AVE | 7.5 | 0.8 | 5.7 | 1.4 | 7.1 | 1.1 | 6.4 | 3.2 |
| | STD | 0.8 | 0.2 | 0.5 | 0.8 | 0.5 | 0.5 | 0.8 | 0.8 |
| | RE (%) | | 89.3 | | 75.5 | | 84.0 | | 49.6 |
| $NH_3$-N (mg L$^{-1}$) | AVE. | 6.0 | 0.1 | 5.2 | 0.0 | 5.2 | 0.2 | 5.0 | 1.2 |
| | STD | 0.9 | 0.1 | 0.6 | 0.0 | 0.3 | 0.3 | 0.3 | 0.9 |
| | RE (%) | | 99.0 | | 100.0 | | 95.8 | | 76.8 |
| $NO_3^-$ -N (mg L$^{-1}$) | AVE. | 1.5 | 0.7 | 0.0 | 1.4 | 1.9 | 0.9 | 1.4 | 2.1 |
| | STD | 0.5 | 0.2 | 0.0 | 0.9 | 0.4 | 0.6 | 0.6 | 0.1 |
| | RE (%) | | 50.6 | | - | | 51.5 | | -46.8 |
| T-P (mg L$^{-1}$) | AVE. | 1.7 | 0.1 | 1.7 | 1.1 | 1.1 | 0.1 | 1.4 | 0.7 |
| | STD | 0.3 | 0.1 | 0.2 | 0.1 | 0.1 | 0.1 | 0.2 | 0.2 |
| | RE (%) | | 94.9 | | 35.5 | | 88.3 | | 49.0 |
| $PO_4^{3-}$ -P (mg L$^{-1}$) | AVE. | 1.2 | 0.1 | 1.3 | 0.9 | 0.8 | 0.1 | 1.0 | 0.6 |
| | STD | 0.2 | 0.1 | 0.2 | 0.1 | 0.1 | 0.1 | 0.1 | 0.2 |
| | RE (%) | | 94.4 | | 31.6 | | 86.8 | | 45.2 |
| MLSS (mg L$^{-1}$) | AVE | 1535.0 | 1800.0 | 1520.0 | 1710.0 | 1460.0 | 1620.0 | 1543.3 | 1631.7 |
| | STD | 115.0 | 50.0 | 49.7 | 29.4 | 40.0 | 20.0 | 33.0 | 35.7 |
| | dX* | | 265.0 | | 190.0 | | 160.0 | | 88.3 |

(Summer: approximately 4 months data (n = 120), winter: approximately 3 months data; n = 90). AVE stands for average, STD stands for standard deviation, and dX* indicates the change of MLSS from the initial to the final reaction time.

MLSS in the profile of AGS: 190 mg L$^{-1}$). Microbial growth that was evident as an increase of MLSS was much higher in summer than in winter, with the removal of $COD_{Cr}$ (eco-HEMS: 110.9 ± 10.4 mg L$^{-1}$ to 40.1 ± 1.7 mg L$^{-1}$, 63.9% RE; AGS: 122.5 ± 9.1 mg L$^{-1}$ to 46.0 ± 9.6 mg L$^{-1}$, 62.5% RE). Interestingly, the trend of N removal in the eco-HEMS profile indicated effective nitrification and denitrification, with a higher RE of T-N compared to that of AGS (eco-

**Table 4. Kinetic parameters' average values obtained from the profile data between AGS and eco-HEMS application at each condition.**

| Season | Sludge | COD:N:P | F/M (g $COD_{Cr}$ • g $MLSS^{-1}$•d$^{-1}$) | Y (g MLVSS • g $COD_{Cr}^{-1}$ •d$^{-1}$) | μ (g• m$^{-3}$d$^{-1}$) | $SV_{30}$ (mL•L$^{-1}$) | SVI (mL•g$^1$) | SSR (g $COD_{Cr}$ removal • g $MLVSS^{-1}$• d$^{-1}$) | SNR (g $NH_3$-N removal • g $MLVSS^{-1}$• d$^{-1}$) | SdNR (g $NO_3^-$-N removal • g $MLVSS^{-1}$• d$^{-1}$) | SPR (g $PO_4^{3-}$-P removal • g $MLVSS^{-1}$• d$^{-1}$) |
|---|---|---|---|---|---|---|---|---|---|---|---|
| Winter | eco-HEMS | 100:5:1 | 0.236 | 0.917 | 0.624 | 80 | 54.8 | 1.091 | 0.1044 | 0.0203 | 0.0136 |
| | | 200:5:1 | 0.406 | 0.534 | 0.802 | 90 | 57.0 | 1.874 | 0.0662 | -0.0109 | 0.0234 |
| | | 300:5:1 | 0.697 | 0.395 | 1.080 | 90 | 58.4 | 2.531 | 0.0831 | -0.0033 | 0.0172 |
| | AGS | 100:5:1 | 0.241 | 0.745 | 0.223 | 200 | 129.6 | 0.894 | 0.0968 | -0.0167 | 0.0117 |
| Summer | eco-HEMS | 100:5:1 | 0.241 | 1.123 | 0.956 | 80 | 52.1 | 0.890 | 0.0742 | 0.0095 | 0.0143 |
| | AGS | 100:5:1 | 0.269 | 0.745 | 0.471 | 150 | 98.7 | 0.895 | 0.0608 | -0.0164 | 0.0047 |

(n = 4 cycles)

HEMS: $7.5 \pm 0.8$ mg $L^{-1}$ to $0.8 \pm 0.2$ mg $L^{-1}$, 89.3% RE; AGS: $5.7 \pm 0.5$ mg $L^{-1}$ to $1.4 \pm 0.8$ mg T-N $L^{-1}$, 75.5% RE). Although the AGS showed a 100% RE of $NH_3$-N, the RE of T-N was lower because concentration of $NO_3^-$-N was high in the effluence (a slight decrease of $NO_3^-$-N was evident from 120 min to 240 min). However, both AGS and eco-HEMS showed effective nitrogen activity in the summer using the same optimal condition. Regarding the removal of P, eco-HEMS displayed a markedly higher RE of P than AGS. The use of eco-HEMS resulted in the prompt removal of T-P and $PO_4^{3-}$-P within the first 30 min, while the use of AGS produced a slight decrease of both T-P and $PO_4^{3-}$-P during SBR cycles.

**1.3. Kinetic parameters in the optimal conditions.** The kinetic parameters were obtained from data collected during the run of the SBR biological treatment system using optimal conditions (Table 4).

The optimal operation conditions (C: N: P = 100: 5: 1; F/M, 0.236–0.241 g $COD_{Cr} \cdot$ g $MLSS^{-1} \cdot d^{-1}$) were as follows: Q, 15.0 $m^{-1}d^{-1}$; $T_C$, 4 h; numbers of cycles per day ($N_C$), 6; HRT, 7.2 h; and SRT, 20.0 d (winter 24.5 d). As the COD ratio increased, the specific growth rate ($\mu$) and SSR increased, but bacterial growth yield (Y) and SNR decreased. The SdNR and SPR were not significantly related. In the N and P removals, the use of eco-HEMS in winter produced higher SNR and SdNR at the C: N: P ratio of 100: 5: 1 than at the other ratios, but SPR was slightly lower at the 100:5:1 ratio than at the other ratios. Thus, a C: N ratio of 20: 1 was the optimal condition for the eco-HEMS-mediated removal of N.

The kinetic parameters of eco-HEMS were much higher than those of AGS. At the optimal 100:5:1 ratio, the obtained kinetic parameters of eco-HEMS in winter were: Y, 0.917 g MLVSS $\cdot$ g $COD_{Cr}^{-1} \cdot d^{-1}$; $\mu$, 0.624 g $m^{-3}$ $d^{-1}$; $SV_{30}$, 80 mL $L^{-1}$; SVI, 54.8 mL $g^{-1}$; SSR, 1.091 g $COD_{Cr}$ removal $\cdot$ g $cell^{-1} \cdot d^{-1}$; SNR, 0.1044 g $NH_3$-N removal $\cdot$ g $cell^{-1} \cdot d^{-1}$; SdNR, 0.0203 g $NO_3^-$ -N removal $\cdot$ g $cell^{-1} \cdot d^{-1}$; and SPR, 0.0136 g $PO_4^{3-}$-P removal $\cdot$ g $cell^{-1} \cdot d^{-1}$. The kinetic parameters of eco-HEMS in the summer were higher than those in the winter: Y, 1.123 g MLVSS $\cdot$ g $COD_{Cr}^{-1} \cdot d^{-1}$; $\mu$, 0.956 g $\cdot$ $m^{-3}$ $d^{-1}$; $SV_{30}$, 80 mL $L^{-1}$, SVI, 52.1 mL $g^{-1}$; SSR, 0.890 g $COD_{Cr}$ removal $\cdot$ g $MLVSS^{-1} \cdot d^{-1}$; SNR, 0.0742 g $NH_3$-N removal $\cdot$ g $MLVSS^{-1} \cdot d^{-1}$; SdNR, 0.0095 g $NO_3^-$ -N removal $\cdot$ g $MLVSS^{-1} \cdot d^{-1}$; and SPR, 0.0143 g $PO_4^{3-}$-P removal $\cdot$ g $MLVSS^{-1} \cdot d^{-1}$. Interestingly, marine wastewaters generally decrease the settling of sludge during wastewater treatment due to high salinity [14]. Presently, the eco-HEMS marine sediment sludge displayed pronounced higher settling with low SVI values, even in the winter season. Subsequent sections in this paper provide further details of why the bacterial community in the eco-HEMS condition resulted in effective N and P removal with higher settling in winter.

# 2. Bacterial community diversity and richness

**2.1. Adaptation and stabilization of marine sludge to eco-HEMS.** As the marine sediment adapted and changed to eco-HEMS in winter, the bacterial community changed and stabilized with stable diversity and richness. Firstly, an analysis of the operational taxonomic units (OTUs) revealed the most diverse with 286 OTUs, 286 Chao1 and 7.57 Shannon's index in the initial marine sediment, with a decrease to 126 OTUs by week 2 when the nitrogen and phosphorus removal bacterial strains applied (Table 5).

The values of OTUs and Chao1 increased again, but Shannon's index decreased from week 3 as the SBR reactor system began operating. As the marine sediment adapted and changed to eco-HEMS, the Shannon index for the abundance and diversity of the bacterial community decreased from 7.57 to 3.04, and the Simpson index decreased from 0.99 to 0.63. These findings indicated the stabilization of the bacterial composition. PC plots revealed relative differences of the bacterial communities during the period of adaptation of the marine sediment to eco-HEMS, with the bacterial communities shifting relatively closer in the S8 Fig

**Table 5. Comparative results of diversity and abundance values (OTUs, Chao1, Shannon, Simpson) from bacteria community analysis.**

| Sludge | Season | Week | OTUs | Chao1 | Shannon | Simpson | Goods Coverage |
|--------|--------|------|------|-------|---------|---------|----------------|
| Eco-HEMS | Winter | 0 | 286 | 286 | 7.57 | 0.99 | 1.00 |
| | | 1 | 133 | 141 | 5.07 | 0.92 | 1.00 |
| | | 2 | 126 | 126 | 4.33 | 0.89 | 1.00 |
| | | 3 | 168 | 174 | 4.65 | 0.92 | 1.00 |
| | | 4 | 183 | 187 | 3.69 | 0.77 | 1.00 |
| | | 5 | 211 | 211 | 3.04 | 0.63 | 1.00 |
| | Summer | | 190 | 211 | 5.42 | 0.94 | 1.00 |

**2. 2. Relative abundance and taxonomic assignment of bacterial community.** In the adaptation period of the marine sediment, the marine sediment changed to effective marine sludge with high performance including nitrogen and phosphorus removal. Bacterial community analysis revealed how effective bacteria (*Bacillus* sp. and *Vibrio* sp.) had dominated in the eco-HEMS during operation. In the phylum level analysis of the bacterial community, phylum Proteobacteria was the most abundant, with the relative abundance increasing to 76.8% in week 4, followed by Bacteroidetes, Planctomycetes, and Firmicutes (Fig 3 and S2 Table).

Phylum Bacteroidetes consisted comprised approximately 20% of the bacterial community of eco-HEMS. In week 1, the relative abundance of phylum Cyanobacteria was 24.4% and phylum Firmicutes was 25.4%.

In more detail, the most abundant 20 genera indicated a change of the eco-HEMS bacterial community, with the most pronounced change being the treatment of effluent from land-based fish farms (Fig 4 and S2 Table).

There was no overwhelmingly predominant genus above 10% of relative abundance in the marine sediment in the beginning (week 0). Thirteen genera displayed relative abundance ranging from 1.1% to 5.6%. After week 1, Genus *Bacillariophyta* was predominant (24.4%), followed by *Thioprofundum* (7.8%), *Robiginitalea* (5.5%), and *Sulfurovum* (4.7%). In week 2, following the application of the high efficiency bacteria to eco-HEMS, the genera of *Vibrio* and *Bacillus* were predominant (relative abundance of 35.3% and 20.1%, respectively). Other dominant genera in terms of relative abundance were *Thalassomonas* (7.8%), *Marinomonas* (5.8%), and *Pseudoalteromonas* (5.0%). In week 3, nitrogen removal bacteria *Vibrio* was still predominant, but the relative abundance had decreased to 22.7%. Another predominant genus was *Psychromonas* (13.9%), followed by *Gaetbulibacter* (9.1%), *Psychroserpens* (5.4%), *Cobetia* (5.0%), and *Bacillus* (5.0%). In week 4, genus *Psychromonas* was predominant with a relative abundance of 45.6%. The genus *Vibrio* was also predominant with a relative abundance of 13.3%, which was a decrease from the week before. Other dominant genera in terms of relative abundance were *Gaetbulibacter* (5.7%) and *Psychroserpens* (4.3%). In week 5, the predominant genus was again *Psychromonas* (60.0%), followed by *Gaetbulibacter* (6.0%), *Glaciecola* (4.7%), and *Psychroserpens* (4.3%). The relative abundance of *Vibrio* sharply decreased to 1.0%. The data indicate that the predominant genera *Psychromonas*, *Vibrio*, *Gaetbulibacter*, and *Psychroserpens* could influence the treatment efficiency of eco-HEMS in winter. Especially, genus *Psychromonas* includes halophilic (high-salt adapted) and psychrophilic (low temperature adapted) species, which display various chemotrophic metabolic activities including nitrification and denitrification, and the synthesis of polyunsaturated fatty acids, such as eicosapentaenoic acid and docosahexaenoic acid [29–31].

In comparison to the bacterial community that developed in summer (S3 Table), phylum *Proteobacteria* was the most abundant in the eco-HEMS bacterial communities, with a relative abundance of 75.9%. Analytical results of the eco-HEMS summer bacterial community

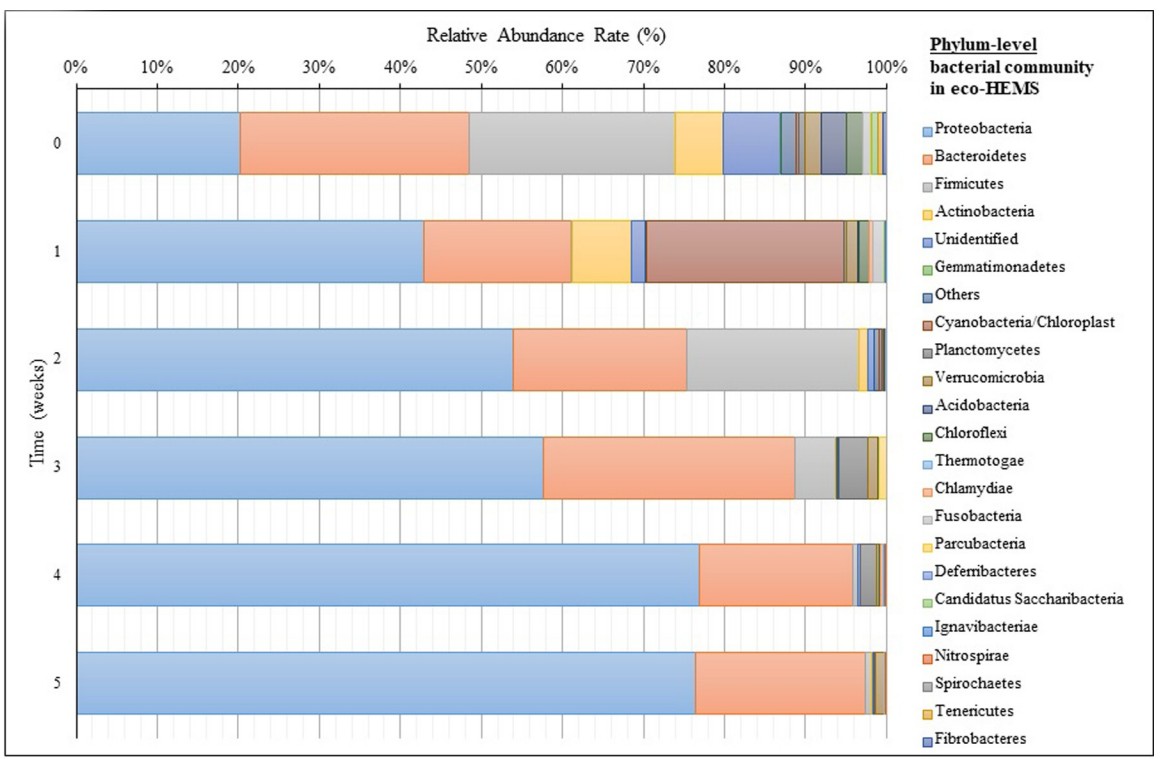

**Fig 3. Analysis of bacterial community at the phylum level during the adaptation of marine sediment to eco-HEMS sludge.**

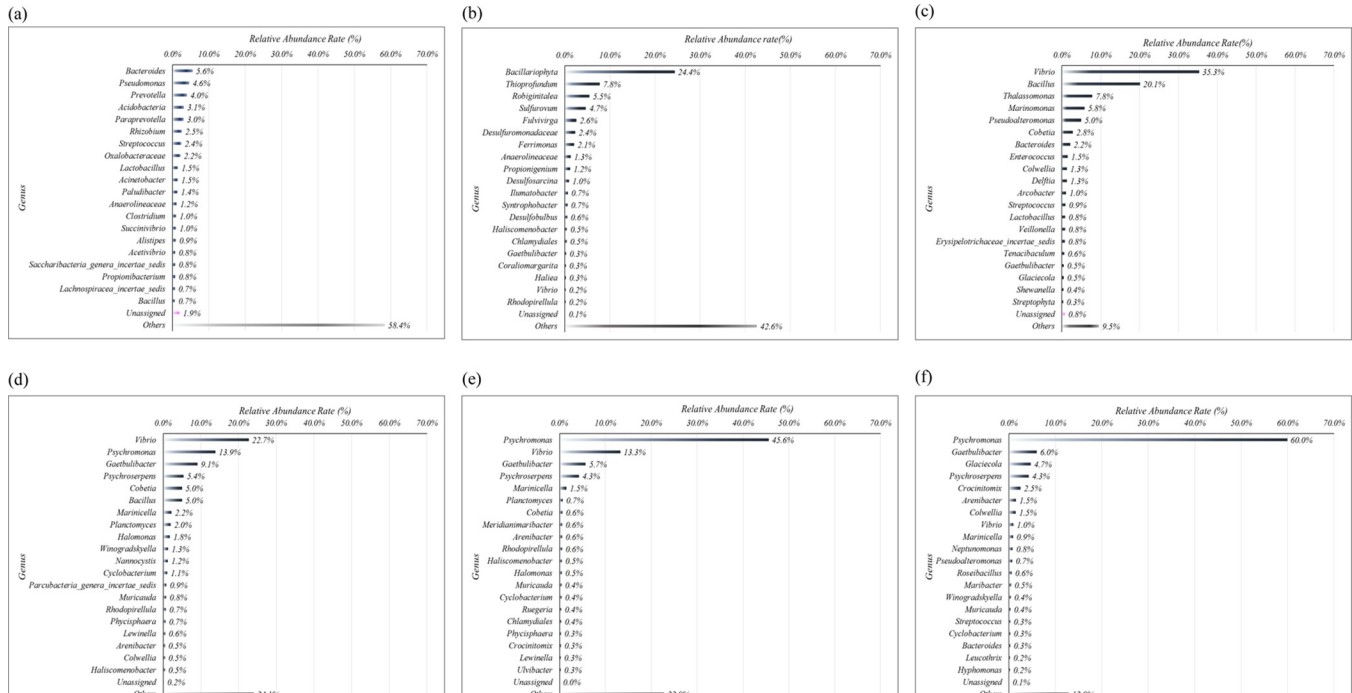

**Fig 4. Top 20 genera concerning abundance rate during the adaptation of marine sediment to marine sludge.** (a) week 0, (b) week 1, (c) week 2, (d) week 3, (e) week 4, and (f) week 5.

revealed the predominant phyla in terms of relative abundance were *Planctomycetes* (9.9%), *Bacteroidetes* (6.3%), *Parcubacteria* (1.3%), and *Chloroflexi* (1.1%). Unlike the bacterial community in winter, 11 of the top 20 genera displayed a relative abundance of more than 1% in the eco-HEMS summer bacterial community. The predominant genera with a relative abundance of more than 10% were *Denitromonas* (17.5%) and *Vibrio* (14.7%). Other genera of family *Rhodobacteraceae* had a relative abundance rate of 10.4%. Dominant genera were family *Phycisphaeracea* (9.7%), genus *Arcobacter* (8.9%), *Roseovarius* (4.5%), *Leucothrix mucor* (2.6%), and *Bacillus* (2.1%). Among them, genus *Denitromonas* is important in the settling of sludge in marine wastewater with pronounced reduction of nitrate [32, 33].

Finally, the bacterial community analyses revealed that the predominant genera in eco-HEMS were different between winter and summer, but the some genera in the marine sediment had mainly adapted to the marine wastewater for efficient removal of N and P. Thus, each predominant genus in summer or winter could be an important influence on the treatment efficiency and settlement. Moreover, the predominant phylum was Proteobacteria. The approximately 70% relative abundance rate of Proteobacteria revealed the presence of stable sludge with high RE in the SBR biological treatment for effluent from the land-based fish farms. Interestingly, the high efficiency bacteria (*Vibrio* and *Bacillus*) did adapt, but did not remain predominant for long-term. The high efficiency bacteria could not be determined how influential to the bacterial community, but bacterial community in eco-HEMS has been quickly stabilized after added those bacterial strains. Otherwise, anammox bacteria did not show in the bacterial community data. Lin *et al*. (2018) described that low temperature inhibited anammox process to treat wastewater [34]. However, denitrification and phosphorus uptake rate were high from the treatment data in optimal condition with above predominant genera, that is, those predominant genera *Psychromonas*, *Vibrio*, *Gaetbulibacter*, and *Psychroserpens* (in winter), *Denitromonas*, *Vibrio*, other genera of family *Rhodobacteraceae*, *Phycisphaeracea*, genus *Arcobacter*, *Roseovarius* (in summer) in eco-HEMS could make role for denitrifying phosphorus removal (DPR). Mandel *et al*. (2018) reported that denitrifying phosphorus removal bacteria adapted and increased in the sludge during biomass adaption [22, 24].

## Conclusions

Eco-HEMS from marine sediment as a biological resource was applied to the SBR biological treatment system in a 2-year pilot plant-scale study for marine wastewater from land-based fish farms. The results demonstrate that eco-HEMS can improve the treatment efficiency of N and P in the high salinity marine wastewater with a daily treatment volume of 15 m$^3$. The treatment efficiency of eco-HEMS is maintained in the winter as well as other seasons. The RE of N is markedly enhanced, as is settlement. The bacterial community appears capable of adapting and optimizing to eco-HEMS during the operation period of the SBR biological treatment. This eco-friendly and economic biological resource could contribute to an effective solution to reduce the cause of HABs by reducing the nutrients presence in the effluent from land-based fish farms or wastewater treatment in coastal areas.

## Supporting information

**S1 Fig.** The locations of marine-sediment sampling site (a) and pilot plant setup (b) in the South Korea (c).
(DOCX)

**S2 Fig. Analytical data of environment factors such as temperature, pH, and salinity in marine wastewater for the pilot plant-scale SBR treatment system during all operation**

**period.**
(DOCX)

**S3 Fig.** Treatment of organic carbon (CODCr, (a)), total nitrogen (b), total phosphorus (c) by the eco-friendly high efficiency marine sludge (eco-HEMS) and the aerobic granule sludge (AGS) in the pilot plant-scale SBR biological treatment system during operation period.
(DOCX)

**S4 Fig. The analytical profile of eco-HEMS applying SBR treatment system in winter when COD: N: P ratio was 200: 5: 1.** (a), environment factors; (b), COD$_{Cr}$ and MLSS, (c), NH$_3$-N, NO$_3^-$ -N, and T-N; (d) PO$_4^{3-}$ -P and T-P. SBR stages consisted of follow 4 stages: I, influence; II, aeration and mixing reaction; III, anaerobic and settlement, IV, decant-effluence and idle.
(DOCX)

**S5 Fig. The analytical profile of eco-HEMS applying SBR treatment system in winter when COD: N: P ratio was 300:5:1.** (a), environment factors; (b), COD$_{Cr}$ and MLSS, (c), NH$_3$-N, NO$_3^-$ -N, and T-N; (d) PO$_4^{3-}$ -P and T-P. SBR stages consisted of follow 4 stages: I, influence; II, aeration and mixing reaction; III, anaerobic and settlement, IV, decant-effluence and idle.
(DOCX)

**S6 Fig. The analytical profile of eco-HEMS applying SBR treatment system in summer when COD: N: P ratio was 100: 5: 1.** (a), environment factors; (b), COD$_{Cr}$ and MLSS, (c), NH$_3$-N, NO$_3^-$ -N, and T-N; (d) PO$_4^{3-}$ -P and T-P. SBR stages consisted of follow 4 stages: I, influence; II, aeration and mixing reaction; III, anaerobic and settlement, IV, decant-effluence and idle.
(DOCX)

**S7 Fig. The analytical profile of AGS applying SBR treatment system in winter when COD: N: P ratio was 100: 5: 1.** (a), environment factors; (b), COD$_{Cr}$ and MLSS, (c), NH$_3$-N, NO$_3^-$ -N, and T-N; (d) PO$_4^{3-}$ -P and T-P. SBR stages consisted of follow 4 stages: I, influence; II, aeration and mixing reaction; III, anaerobic and settlement, IV, decant-effluence and idle.
(DOCX)

**S8 Fig. The Principle Component Analysis (PCA) with PC plots for bacterial communities during adaptation period from the marine sludge to eco-HEMS. (a)** PCA analysis based on PC1 vs PC2, **(b)** PCA analysis based on PC1 vs PC3, (c) PCA analysis based on PC3 vs PC2; 0 week (●), 1 week (■), 2 week(▶), 3week (▲),4week ◀), 5 week (▼).
(DOCX)

**S1 Table. Comparative analytical data of COD$_{Cr}$, T-N, T-P between eco-friendly high efficiency marine sludge (eco-HEMS) and aerobic granule sludge (AGS) application in pilot plant-scale SBR treatment system during all operation period.**
(DOCX)

**S2 Table. Comparative results of diversity and abundance values (OTUs, Chao1, Shannon, Simpson) from bacteria community analysis.**
(XLSX)

**S3 Table. Comparative analysis of bacterial community between eco-HEMS and AGS in summer.**
(XLSX)

## Author Contributions

**Conceptualization:** Jinsoo Kim.

**Data curation:** Jinsoo Kim.

**Formal analysis:** Jinsoo Kim.

**Funding acquisition:** Sang-Seob Lee.

**Investigation:** Jinsoo Kim, Sangrim Kang, Hyun-Sook Kim, Sungchul Kim.

**Methodology:** Jinsoo Kim, Sangrim Kang, Hyun-Sook Kim, Sungchul Kim.

**Project administration:** Sangrim Kang, Sungchul Kim.

**Supervision:** Sang-Seob Lee.

**Writing – original draft:** Jinsoo Kim.

**Writing – review & editing:** Jinsoo Kim.

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
