## [Decision Letter · Decision Letter 0]

19 Nov 2019

PONE-D-19-18032

Pilot Plant Study on Nitrogen and Phosphorus Removal in Marine Wastewater by Marine Sediment with Sequencing Batch Reactor

PLOS ONE

Dear Professor Lee,

Thank you for submitting your manuscript to PLOS ONE. After careful consideration, we feel that it has merit but does not fully meet PLOS ONE’s publication criteria as it currently stands. Therefore, we invite you to submit a revised version of the manuscript that addresses the points raised during the review process.

We would appreciate receiving your revised manuscript by Jan 03 2020 11:59PM. To enhance the reproducibility of your results, we recommend that if applicable you deposit your laboratory protocols in protocols.io, where a protocol can be assigned its own identifier (DOI) such that it can be cited independently in the future. For instructions see: http://journals.plos.org/plosone/s/submission-guidelines#loc-laboratory-protocols

We look forward to receiving your revised manuscript.

Kind regards,

Arumugam Sundaramanickam, PhD

Academic Editor

PLOS ONE

Journal Requirements:

1. Thank you for including your competing interests statement; "The authors have declared that no competing interests exist."

We note that one or more of the authors are employed by a commercial company:

"Research & Development Institute of Inventory Co. Ltd"

Additional Editor Comments (if provided):

Dear Sang-Seob Lee,

Thank you for sending us your article and giving us the chance to consider your work. Your article was carefully reviewed by external reviewers for the consideration of publication in PLOS ONE. Unfortunately, the manuscript was not able to accept in the present form. However, we consider your article after complete revision of your manuscript. You are advised to address the comments raised by the reviewers and revise your manuscript. I strongly encourage you to seek the support of a native English speaker language expert and advised to make a new version of your manuscript.

Reviewers' comments:

Reviewer's Responses to Questions

**Comments to the Author**

1. Is the manuscript technically sound, and do the data support the conclusions?

Reviewer #1: Yes

Reviewer #2: Yes

Reviewer #3: No

2. Has the statistical analysis been performed appropriately and rigorously? 

Reviewer #1: I Don't Know

Reviewer #2: Yes

Reviewer #3: No

3. Have the authors made all data underlying the findings in their manuscript fully available?

Reviewer #1: Yes

Reviewer #2: Yes

Reviewer #3: Yes

4. Is the manuscript presented in an intelligible fashion and written in standard English?

Reviewer #1: No

Reviewer #2: Yes

Reviewer #3: No

5. Review Comments to the Author

Reviewer #1: This article reports on the results of a pilot plant study that treats agricultural marine wastewater via a sequencing batch reactor. This seems to be the follow up to a previous laboratory scale SBR study. Overall, I find the study to be of high interest and overall it was well done. The manuscript is organized well. The main problem I see is that the authors are not specific with regards to what was done in replicate analysis. I strongly advise that the authors be clear in the methods with what kind of replicates was done.

The authors should be more clear in the introduction or abstract that this builds on their previous study.

I am not a fan of the ‘eco’ part of the ‘eco-HEMS’ name – it seems a little bit like salesmanship and all wastewater treatment can be called ‘eco-friendly’. However, since the term was used in prior studies, the authors could also consider keeping it for consistency purposes.

Though I had no problems understanding the authors, there are many areas in which grammar could be improved (I point to some cases in details below), especially in the abstract. The authors and should carefully review the final version for grammar, especially since PLOSONE won’t help. I hope the authors find my comments on some of these grammar errors are helpful.

I think the authors could be more specific with how they determined “optimal operational conditions”. What was the basis of ‘optimal’, and what variables were tested?

Specific comments:

Line 2. Is it more difficult because of the salinity or do we just know less about it. I think it is a reach to say it is difficult, since whole oceans are out there with all sorts of P and N cycling processes being done on a massive scale. I recommend the authors stating something more like ”Effective biological treatment of marine wastewater is not well-known.”

Line 2-3. The grammar on this sentence is odd. “Accumulation of nitrogen and phosphorus from land-based effluent is a crucial cause of red-tide in marine systems” makes more sense.

Line 4: Add “The” before purpose

Line 5. “a” instead of “the” before “pilot plant-scale sequencing…”

Line 6. After the common, edit to “ elucidate which bacterial strains in sludge from marine sediment influence the performance of the SBR”.

Line 9: “Improved by” doesn’t really make sense, maybe delete it?

Line 11. After summer, use a period and start a new sentence with “moreover”.

Line 12: “Unlike the common treatment” would be better grammar, and what is the common treatment? Please specify here so that the reader knows the main point of reference

Line 16: This isn’t specific enough, please state the method. (i.e. state something like “From 16S rRNA amplicon sequencing, the predominant genera was found to be ….” ). Too often, scientists are becoming careless with regards to what high throughput sequencing data gives us. If there is a PCR amplification, you don’t have a ‘true’ relative abundance, what you have a messy estimate. Thus, it needs to be clearer when we scientists are getting composition estimates based on sequencing of amplicons, vs when we are using a method that gives us a much better estimate of relative abundance (like from total rRNA direct sequencing, or true metagenomics sequencing, etc).

Line 28. Inland aquaculture and fish production may have cause HAB in some marine coastal regions, but the way this is stated grammatically implied is the main cause, which I don’t think is true.

Line 33: This needs revised, it reads that government and scientists are being monitored! A suggested edit would be “In recent decades, scientific and government monitoring in South Korea have found that increasing…”

Line 36: A simplification would be “important in a long-term prevention strategy.”

Line 37. This sentence should be edited as well. “Treatment of marine wastewater needs better understood because the …..”

Line 60: Suggested edit “…to study possible application of …”

Line 61. I suggest deleting “more”

Line 71: use “a” instead of “the” to refer to the land based fish farm.

Line 72: Replace “with a” instead of “and the” before pilot plant SBR. I think everything in that sentence after SBR can be deleted.

Line 74. Put a period after (eco-HEMS), and start “we” as a new sentence.

Line 75. “in bacterial community” is redundant information and can be deleted.

Line 76. Delete ”and predominated”

Line 86 and 88. Instead of “is efficient remove”, just write “efficiently removes”

Lind 107. “Was” instead of “ware” and the rest of this sentence needs some help (the MLSS was chosen based on the lab scale reactors?)

Line 180. “as the control” doesn’t make sense to me, I would just delete these words.

Line 326. “stable” can be deleted

Line 340. This sentence is confusing.

Line 343. Instead of “effective bacteria”, can you use a more specific term (i.e. genera names? Are these the added bacteria?

Line 349. Is this also week 1?

Line 350. The plural of genus is genera.

How many reads were gathered in these samples? The read depth should be stated somewhere.

Line 389-390. This sentence is confusing and needs edited.

Line 398. Not sure how anammox connects to the information preceeding this sentence. The Anammox bacteria are Planctomycetes, which was observed (as reported in lines 379 – 9.9% Planctomycetes is actually relatively high for a non anammox reactor). Thus, I am not sure why they also say that they did not shown in the bacterial community data, unless these Planctomycetes are all from classes not known to have anammox, in which case, this should be better stated.

Reviewer #2: The baits in mariculture industry bring a lot of nutrients to the offshore area and destroy the balance of the marine ecosystem, causing red tide and other phenomena. An efficient device has been developed in this study to deal with nutrients such as N and P. This study has a high application value. The research methods and results of this paper are very detailed and logical. I think it is suitable for publication in this journal. There is only one suggestion, i.e. almost all the text in “results and discussion” is only “results”, there is almost no “discussion”. For example, what causes seasonal changes in microbial diversity? How tolerant are these strains to the different environment? I think the readers are also very interested in these mechanisms. However, mechanisms may be difficult to explain because it is not the focus of this research. Anyway, it is worthy to make more discussion.

Reviewer #3: Summary of the manuscript

In this paper the overall idea was really good, but the methods were doesn’t have clarity and not well organized. Manuscript has lots of short comings and more related discussions should be included. It can’t be considered for the publication in the same format.

Specific corrections

The manuscript needs an extensive review of the English language, manuscript format and also spelling mistakes.

Authors have mentioned that 89.3% & 94.9% removal during summer and 84% & 88% during winter, but stated that higher removal observed during winter. Avoid controversy statements.

Avoid earlier study’s results discussions in the methodology part (line no. 89-93); it can be included in the introduction or discussion part.

The authors stated that they used the bacterial strains of Bacillus sp. KGN1and Vibrio sp. KGP1 for the treatment of TN and TP. Since, these strains are very potential why the authors are not taken any effort to confirm the species?

Supporting evidence for the seasonal fluctuation of environmental parameters is missing. Parameters such as Temperature, Salinity and pH are closely related parameter but in the present study, authors have not observed any correlation between the parameters.

The study can be improved by using statistical tools

The salinity high during winter and was decreased during spring and summer, how it possible?

Discussion part to be improved

The references are not in the correct format; lots of punctuations errors are noticed.

6. PLOS authors have the option to publish the peer review history of their article (what does this mean?). If published, this will include your full peer review and any attached files.

Reviewer #1: No

Reviewer #2: No

Reviewer #3: No

---

## [Author Response · Author response to Decision Letter 0]

21 Mar 2020

Dear Dr. Arumugam Sundaramanickam,

Thank you for giving me the opportunity to submit a revised draft of my manuscript titled “ PONE-D-19-18032: Pilot plant study on nitrogen and phosphorus removal in marine wastewater by marine sediment with sequencing batch reactor” to PLOS ONE. We appreciate the time and effort that you and the reviewers have dedicated to providing your valuable feedback on our manuscript. We are grateful to the reviewers for their insightful comments on our paper. We have been able to incorporate changes to reflect most of the suggestions provided by the reviewers. We have attached followings: your letter and a point-by point response to the reviewers’ comments and concerns.

Sincerely, 

Sang-Seob Lee

Professor, 

Department of Life Sciences,

College of Convergence and Integrated Science

Kyonggi University

PLOS ONE

Journal Requirements:

Response: Thank you for your comments information. Therefore, we revised the manuscript as PLOS ONE’s style.

1. Thank you for including your competing interests statement; "The authors have declared that no competing interests exist."

We note that one or more of the authors are employed by a commercial company:

"Research & Development Institute of Inventory Co. Ltd"

Response: Thank you for your comments. ‘R&D institute of Inventory Co. Ltd’ did not play a role with any funding. Also, The authors, Jinsoo Kim and Sungchul Kim, are not employed by Inventory Co. Ltd any more. Thus, we decided to erase the affiliation ‘R&D institute of Inventory Co. Ltd’ but remained the affiliation ‘Kyonggi university’. We are not related to the following comments “1, 2”. 

Response: Thank you for your comments and we included captions for supporting information

files at the manuscript and update in-text citations to match accordingly as you commented and gave information. 

Response: Thank you for your comments and the corresponding author ‘Sang-Seob Lee’ has updated information with ORCID ID. 

 

Additional Editor Comments (if provided):

Dear Sang-Seob Lee,

Thank you for sending us your article and giving us the chance to consider your work. Your article was carefully reviewed by external reviewers for the consideration of publication in PLOS ONE. Unfortunately, the manuscript was not able to accept in the present form. However, we consider your article after complete revision of your manuscript. You are advised to address the comments raised by the reviewers and revise your manuscript. I strongly encourage you to seek the support of a native English speaker language expert and advised to make a new version of your manuscript.

Response: Thank you for your review. We revised the manuscript as reviewer’s comments, and we have completed the revision of the manuscript with correction and improving English. We also have taken an English editing service from Editage company, which is a recommended English editing company by PLOSONE.

 

Reviewers' comments:

Reviewer's Responses to Questions

Comments to the Author

1. Is the manuscript technically sound, and do the data support the conclusions?

Reviewer #1: Yes

Reviewer #2: Yes

Reviewer #3: No

Response: Thank you for your review. We revised the conclusion from obtained data in the study. Additionally, we repeatedly performed and analyzed in every day and every cycle with more than 2 weeks at the same operational condition of the pilot plant SBR system.

2. Has the statistical analysis been performed appropriately and rigorously?

Reviewer #1: I Don't Know

Reviewer #2: Yes

Reviewer #3: No

Response: Thank you for your responses. We re-checked the statistical analysis in the study and revised the manuscript.

3. Have the authors made all data underlying the findings in their manuscript fully available?

Reviewer #1: Yes

Reviewer #2: Yes

Reviewer #3: Yes

 Response: Thank you for your review.

4. Is the manuscript presented in an intelligible fashion and written in standard English?

Reviewer #1: No

Reviewer #2: Yes

Reviewer #3: No

Response: Thank you for your review. We revised the manuscript as reviewer’s comments, and we have completed the revision of the manuscript with correction and improving English. We also have taken an English editing service from Editage company, which is a recommended English editing company by PLOSONE.

5. Review Comments to the Author

Reviewer #1: This article reports on the results of a pilot plant study that treats agricultural marine wastewater via a sequencing batch reactor. This seems to be the follow up to a previous laboratory scale SBR study. Overall, I find the study to be of high interest and overall it was well done. The manuscript is organized well. The main problem I see is that the authors are not specific with regards to what was done in replicate analysis. I strongly advise that the authors be clear in the methods with what kind of replicates was done.

Response: Thank you for your review. We repeatedly performed and analyzed in every day and every cycle with more than 2 weeks at the same operational condition of the pilot plant SBR system. Moreover, we commented the replicates for the experiment data as well as methods in detail.

The authors should be more clear in the introduction or abstract that this builds on their previous study.

- Thank you for your comments, we make clear in the introduction or abstract about the previous study. 

I am not a fan of the ‘eco’ part of the ‘eco-HEMS’ name – it seems a little bit like salesmanship and all wastewater treatment can be called ‘eco-friendly’. However, since the term was used in prior studies, the authors could also consider keeping it for consistency purposes.

- Thank you for your comments, indeed, ‘eco-HEMS’ does not imply as commercial purpose, but it emphasizes the meaning of ‘eco-friendly’ and ‘high efficiency’. We used marine sediment and high efficiency bacteria in the previous study (lab-scale study), so we wanted to use both meaning and abbreviate the words, ‘eco-friendly high efficiency marine sludge (eco-HEMS). 

Though I had no problems understanding the authors, there are many areas in which grammar could be improved (I point to some cases in details below), especially in the abstract. The authors and should carefully review the final version for grammar, especially since PLOSONE won’t help. I hope the authors find my comments on some of these grammar errors are helpful.

- Thank you for your kind comments, we revised the manuscript as your specific comments. Indeed, we had confirmed English by Editage company, which is a recommended English editing company by PLOSONE. 

I think the authors could be more specific with how they determined “optimal operational conditions”. What was the basis of ‘optimal’, and what variables were tested?

- Thank you for your kind comments, we searched the operational condition of SBR system for high removal efficiency of N, P. Indeed, we already obtained the optimal conditions about variables (Aerobic/anaerobic time, Q, Tc, HRT, SRT, COD:N:P ratio, F/M, MLSS/MLVSS concentration, and so on) in the previous study (lab-scale reactor), and the data were not shown in this paper. The variables of optimal operational conditions were showed in the Table 1. We repeatedly performed in every operational conditions. 

Specific comments:

Line 2. Is it more difficult because of the salinity or do we just know less about it. I think it is a reach to say it is difficult, since whole oceans are out there with all sorts of P and N cycling processes being done on a massive scale. I recommend the authors stating something more like ”Effective biological treatment of marine wastewater is not well-known.”

-> I agree with you and thank you for your comments. 

We revised the sentence as you recommended. 

Line 2-3. The grammar on this sentence is odd. “Accumulation of nitrogen and phosphorus from land-based effluent is a crucial cause of red-tide in marine systems” makes more sense.

-> I learned from you so much.

Thank you for your comments and we revised the sentence as you recommended. 

Line 4: Add “The” before purpose

-> Thank you for your comments and we revised the sentence as you recommended. 

Line 5. “a” instead of “the” before “pilot plant-scale sequencing…”

-> Thank you for your comments and we revised the sentence as you recommended. 

Line 6. After the common, edit to “ elucidate which bacterial strains in sludge from marine sediment influence the performance of the SBR”.

-> Thank you for your comments and we revised the sentence as you recommended. 

Line 9: “Improved by” doesn’t really make sense, maybe delete it?

-> Thank you for your comments and we delete “improved by” as you recommended. 

Line 11. After summer, use a period and start a new sentence with “moreover”.

-> Thank you for your comments and we revised it as you recommended. 

Line 12: “Unlike the common treatment” would be better grammar, and what is the common treatment? Please specify here so that the reader knows the main point of reference.

-> Thank you for your comments and we revised the sentence as you mentioned. Biological treatment efficiency is lower in winter because general bacteria activity is lower at the lower temperature.

Line 16: This isn’t specific enough, please state the method. (i.e. state something like “From 16S rRNA amplicon sequencing, the predominant genera was found to be ….” ). Too often, scientists are becoming careless with regards to what high throughput sequencing data gives us. If there is a PCR amplification, you don’t have a ‘true’ relative abundance, what you have a messy estimate. Thus, it needs to be clearer when we scientists are getting composition estimates based on sequencing of amplicons, vs when we are using a method that gives us a much better estimate of relative abundance (like from total rRNA direct sequencing, or true metagenomics sequencing, etc).

-> I agree with you, thank you for your comments and we revised the sentence as you mentioned.

Line 28. Inland aquaculture and fish production may have cause HAB in some marine coastal regions, but the way this is stated grammatically implied is the main cause, which I don’t think is true.

-> Thank you for your comments and we revised the sentence. 

Line 33: This needs revised, it reads that government and scientists are being monitored! A suggested edit would be “In recent decades, scientific and government monitoring in South Korea have found that increasing…”

-> Thank you for your comments and we revised the sentence as you suggested. 

Line 36: A simplification would be “important in a long-term prevention strategy.”

-> Thank you for your comments and we revised the sentence as you mentioned. 

Line 37. This sentence should be edited as well. “Treatment of marine wastewater needs better understood because the …..”

-> Thank you for your comments and we revised the sentence as you mentioned. 

Line 60: Suggested edit “…to study possible application of …”

-> Thank you for your comments and we revised the sentence as you mentioned. 

Line 61. I suggest deleting “more”

-> Thank you for your comments and we deleted it as you mentioned. 

Line 71: use “a” instead of “the” to refer to the land based fish farm.

-> Thank you for your comments and we revised it as you mentioned. 

Line 72: Replace “with a” instead of “and the” before pilot plant SBR. I think everything in that sentence after SBR can be deleted.

-> Thank you for your comments and we revised it as you mentioned. 

Line 74. Put a period after (eco-HEMS), and start “we” as a new sentence.

-> Thank you for your comments and we revised it as you mentioned. 

Line 75. “in bacterial community” is redundant information and can be deleted.

-> Thank you for your comments and we deleted it as you mentioned. 

Line 76. Delete ”and predominated”

-> Thank you for your comments and we deleted the sentence as you mentioned. 

Line 86 and 88. Instead of “is efficient remove”, just write “efficiently removes”

-> Thank you for your comments and we revised it as you mentioned. 

Lind 107. “Was” instead of “ware” and the rest of this sentence needs some help (the MLSS was chosen based on the lab scale reactors?)

-> Thank you for your comments and we revised it as you mentioned. 1,500 mg L-1 of MLSS showed the highest efficiencies in the previous lab-scale reactors study.

Line 180. “as the control” doesn’t make sense to me, I would just delete these words.

-> Thank you for your comments and we deleted it as you mentioned. 

Line 326. “stable” can be deleted

-> Thank you for your comments and we deleted it as you mentioned. 

Line 340. This sentence is confusing.

-> Thank you for your comments and we revised the sentence as you mentioned. 

Line 343. Instead of “effective bacteria”, can you use a more specific term (i.e. genera names? Are these the added bacteria?

-> Thank you for your comments and we added bacteria names (Bacillus sp. and Vibrio sp.). 

 Yes, we added these bacteria for improving efficiency. 

Line 349. Is this also week 1?

-> Yes, the result was also week 1.

Line 350. The plural of genus is genera.

-> Thank you for your comments and we revised it as you mentioned. 

How many reads were gathered in these samples? The read depth should be stated somewhere.

Line 389-390. This sentence is confusing and needs edited.

-> Thank you for your comments and we revised it. 

Line 398. Not sure how anammox connects to the information preceeding this sentence. The Anammox bacteria are Planctomycetes, which was observed (as reported in lines 379 – 9.9% Planctomycetes is actually relatively high for a non anammox reactor). Thus, I am not sure why they also say that they did not shown in the bacterial community data, unless these Planctomycetes are all from classes not known to have anammox, in which case, this should be better stated.

-> Thank you for your comments and we re-stated the results and discussion.

 

Reviewer #2: The baits in mariculture industry bring a lot of nutrients to the offshore area and destroy the balance of the marine ecosystem, causing red tide and other phenomena. An efficient device has been developed in this study to deal with nutrients such as N and P. This study has a high application value. The research methods and results of this paper are very detailed and logical. I think it is suitable for publication in this journal. There is only one suggestion, i.e. almost all the text in “results and discussion” is only “results”, there is almost no “discussion”. For example, what causes seasonal changes in microbial diversity? How tolerant are these strains to the different environment? I think the readers are also very interested in these mechanisms. However, mechanisms may be difficult to explain because it is not the focus of this research. Anyway, it is worthy to make more discussion.

-> Thank you so much for your comments. We added more discussion including your comments as your brilliant comment.

 

Reviewer #3: Summary of the manuscript

In this paper the overall idea was really good, but the methods were doesn’t have clarity and not well organized. Manuscript has lots of short comings and more related discussions should be included. It can’t be considered for the publication in the same format.

-> Thank you so much for your comments. We revised the methods and discussion as your brilliant comment.

Specific corrections

The manuscript needs an extensive review of the English language, manuscript format and also spelling mistakes.

-> Thank you for your comments and we revised and checked many mistakes and English problems. 

Authors have mentioned that 89.3% & 94.9% removal during summer and 84% & 88% during winter, but stated that higher removal observed during winter. Avoid controversy statements.

-> Thank you for your comments and we revised it.

Avoid earlier study’s results discussions in the methodology part (line no. 89-93); it can be included in the introduction or discussion part.

-> Thank you for your comments and we included the previous methodology and results into the introduction as you commented. 

The authors stated that they used the bacterial strains of Bacillus sp. KGN1and Vibrio sp. KGP1 for the treatment of TN and TP. Since, these strains are very potential why the authors are not taken any effort to confirm the species?

-> Thank you for your comments. Indeed, we already identified the strains in the earlier previous study.

Supporting evidence for the seasonal fluctuation of environmental parameters is missing Parameters such as Temperature, Salinity and pH are closely related parameter but in the present study, authors have not observed any correlation between the parameters. The study can be improved by using statistical tools

The salinity high during winter and was decreased during spring and summer, how it possible?

Discussion part to be improved

-> Thank you for your comments. Salinity values were decreased due to the typhoon and rainy season occurs in the summer and early autumn on the south coastal area of South Korea. We revised and improved the environmental parameters’ part as you commented.

The references are not in the correct format; lots of punctuations errors are noticed.

 -> Thank you for your comments. We corrected the references as you commented.

---

## [Decision Letter · Decision Letter 1]

28 Apr 2020

Pilot plant study on nitrogen and phosphorus removal in marine wastewater by marine sediment with sequencing batch reactor

PONE-D-19-18032R1

Dear Dr. Lee,

We are pleased to inform you that your manuscript has been judged scientifically suitable for publication and will be formally accepted for publication once it complies with all outstanding technical requirements.

With kind regards,

Arumugam Sundaramanickam, PhD

Academic Editor

PLOS ONE

Additional Editor Comments (optional):

Dear Dr Sang-Seob Lee,

I am pleased to inform you that the reviewers now recommend  your manuscript for publication in PLOS ONE.

Thank you for submitting your work to this journal.

With regards

A. Sundaramanickam

Academic Editor

PLOS ONE

Reviewers' comments:

Reviewer's Responses to Questions

**Comments to the Author**

1. If the authors have adequately addressed your comments raised in a previous round of review and you feel that this manuscript is now acceptable for publication, you may indicate that here to bypass the “Comments to the Author” section, enter your conflict of interest statement in the “Confidential to Editor” section, and submit your "Accept" recommendation.

Reviewer #1: All comments have been addressed

Reviewer #3: All comments have been addressed

Reviewer #4: All comments have been addressed

2. Is the manuscript technically sound, and do the data support the conclusions?

Reviewer #1: Yes

Reviewer #3: Yes

Reviewer #4: (No Response)

3. Has the statistical analysis been performed appropriately and rigorously? 

Reviewer #1: Yes

Reviewer #3: Yes

Reviewer #4: (No Response)

4. Have the authors made all data underlying the findings in their manuscript fully available?

Reviewer #1: Yes

Reviewer #3: Yes

Reviewer #4: (No Response)

5. Is the manuscript presented in an intelligible fashion and written in standard English?

Reviewer #1: Yes

Reviewer #3: Yes

Reviewer #4: (No Response)

6. Review Comments to the Author

Reviewer #1: The authors improved upon their previous submission and appears to have addressed previous reviewer comments. I am satisfied with the revisions and believe it is suitable for publication. It is an interesting area of study and appreciate the authors work studying and developing wastewater treatment processes in this niche area.

Reviewer #3: Comments

Overall the manuscript has revised well, all the comments were addressed in the nice manner. So that I would recommend this manuscript for the acceptance.

Specifically, author has corrected grammatical and spelling errors.

Introduction had improved with previous study results.

Author clarified all the questions and doubts which were raised by me.

References also revised in the same format as I mentioned.

Reviewer #4: (No Response)

7. PLOS authors have the option to publish the peer review history of their article (what does this mean?). If published, this will include your full peer review and any attached files.

Reviewer #1: No

Reviewer #3: No

Reviewer #4: No

---

## [Editor Report · Acceptance letter]

4 May 2020

PONE-D-19-18032R1 

Pilot plant study on nitrogen and phosphorus removal in marine wastewater by marine sediment with sequencing batch reactor 

Dear Dr. Lee:

I am pleased to inform you that your manuscript has been deemed suitable for publication in PLOS ONE. Congratulations! Your manuscript is now with our production department. 

With kind regards,

on behalf of

Professor Arumugam Sundaramanickam 

Academic Editor

PLOS ONE